# Fluorine-Free and Transparent Superhydrophobic Coating with Enhanced Anti-Icing and Anti-Frosting Performance by Using D26 and KH560 as Coupling Agents

Ting Xiao [1,2,3,*], Yudian Wang [1] , Xia Lang [2], Siyu Chen [2], Lihua Jiang [2], Fujun Tao [4], Yequan Xiao [2], Xinyi Li [2] and Xinyu Tan [2,3,*]

1   Hubei Provincial Engineering Technology Research Center for Microgrid, College of Electrical Engineering & New Energy, China Three Gorges University, Yichang 443002, China; wangyudiandian@sina.cn

2   Key Laboratory of Inorganic Nonmetallic Crystalline and Energy Conversion Materials, College of Materials and Chemical Engineering, China Three Gorges University, Yichang 443002, China; 17383038405@163.com (X.L.); c1906090128@163.com (S.C.); jlihua107@ctgu.edu.cn (L.J.); xiaoyequan@ctgu.edu.cn (Y.X.); lixinyi@ctgu.edu.cn (X.L.)

3   Solar Energy High Value Utilization and Green Conversion Hubei Provincial Engineering Research Center, Yichang 443002, China

4   Department of Chemistry and Biochemistry, Northern Illinois University, DeKalb, IL 60115, USA; ftao@niu.edu

*   Correspondence: tingxiao@ctgu.edu.cn (T.X.); tanxin@ctgu.edu.cn (X.T.)

**Abstract:** Superhydrophobic surfaces with non-wetting characteristics have been considered to be potential candidates for ice/frost prevention. In this study, a transparent superhydrophobic coating was created by using a simple method that employed (3-glycidoxypropyl) trimethoxysilane (KH560) and 1,2-Bis (trimethoxysilyl) ethane (D26) as coupling agents and epoxy resin (E51) as an adhesive. The synergy between KH560 and D26 significantly improves the long-term outdoor durability, anti-icing, and anti-frosting performance of the superhydrophobic coating. The coating also has good acid and alkali resistance, UV resistance, and durability. The obtained $SiO_2$@E51@KH560@D26 can delay the freezing time of water by 1974 s, much longer than bare glass (345 s) and also longer than the coatings with only D26 (932 s) or with only KH560 (1087 s). Moreover, the $SiO_2$@E51@KH560@D26 showed an improved anti-frosting capability compared with the other three samples and better maintained its superhydrophobic properties at low temperatures. Our study proposes a potential method to fabricate a superhydrophobic coating with both anti-icing and anti-frosting properties.

**Keywords:** anti-icing; anti-frosting; KH560; D26; coupling agents

## 1. Introduction

The formations of ice and frost in low-temperature environments are common natural phenomena that widely exist in the fields of power transmission, power generation, aerospace, transportation, and so on. They can decrease the operating efficiency of equipment, and in severe cases, even threaten people's lives and property [1–4]. Therefore, attempts at inhibiting icing and frosting on exposed surfaces with various methods have been extensively made. Traditional deicing/frosting technologies mainly include liquid anti-icing, electric anti-icing, gas-heated anti-icing, mechanical deicing, and so on. However, these methods suffer from many problems such as high energy consumption, environmental pollution, low accessibility, and the short duration of the effect. To address these issues, engineering anti-icing/frosting surfaces that can prevent or delay ice/frost accumulation and icephobic surfaces on which ice can be easily removed owing to the low ice adhesion strength have attracted extensive interest [5–7].

Icing/frosting on a solid surface generally begins with the wetting of supercooled water droplets in the environment at a low temperature (in addition, if the temperature is low enough, the frost layer will be directly converted from gaseous water to the solid state, which is not discussed here). Therefore, the non-wetting characteristic is required to prevent icing/frosting on a solid surface. Superhydrophobic surfaces with water contact angles greater than 150° and water sliding angles less than 10°, which show excellent non-wettability, have been considered to be potential candidates for ice/frost prevention. It is well known that both chemical composition and surface structure have an important influence on the wettability of material. The chemical composition can adjust the surface energy of the material, and the surface structure can improve wettability by forming a multi-layer structure and adjusting the hierarchical structure [8]. In terms of anti-icing and anti-frosting, these features can remove water droplets during the condensation stage and reduce the heat conduction efficiency, thus delaying ice/frost nucleation and growth [9,10]. Although various methods, such as electrospinning [11], the template method [12], chemical etching [13], chemical vapor deposition [14], electrodeposition [15], and sol–gel [16], have been used to construct superhydrophobic surfaces, employing simple and inexpensive coating equipment with a low cost and good shape adaptability is of great importance when taking into account the large-scale production capability. Meanwhile, to reduce the surface energy, fluorine-containing compounds are usually used in the preparation of superhydrophobic coatings, which have been shown to have negative effects on the ecological environment and human health [17–20].

In some cases, such as the surface of windows, solar photovoltaic panels, and glass insulators, the superhydrophobic coatings should be transparent. However, the light transmission of the superhydrophobic coating is usually contradictory to the roughness generated by the micro/nanostructures, and an optimal balance should be carefully regulated in this respect [21,22]. Achieving transparent and superhydrophobic coatings utilizing fluorine-free chemical reagents and substrate-independent processes remains a significant difficulty. Furthermore, the low mechanical stability of artificially produced superhydrophobic materials caused by the poor coating/substrate interfacial adhesion strength and frail micro/nanostructures usually hampers the wide use of superhydrophobic coatings. More importantly, during the repeated icing–deicing process, superhydrophobic coatings can lose hydrophobicity easily because of the destruction of the micro/nanostructures. In order to improve the mechanical properties of coatings, the introduction of proper coupling agents has been commonly used. Dong et al. used 3-aminopropyltriethoxysilane (KH550) to modify $SiO_2$ nanoparticles by using a low temperature sol–gel method [23]. A. Bake et al. mixed methyltrimethoxysilane (MTMS) and (3-glycidoxypropyl) trimethoxysilane (KH560) to bond functionalized silica nanoparticles to various substrates and improve their firmness [24]. Tan et al. mixed γ-methacryloxypropyl trimethoxy silane (KH570) with methyl MQ silicone resin and $SiO_2$ nanoparticles to construct superhydrophobic surfaces [25]. Zhu et al. used isocyanate siloxan-3-isopropyl triethoxy-isocyanate (ICPTES) as a coupling agent to treat a new superhydrophobic melamine formaldehyde (MF) sponge by a urea crosslinking process [26]. However, there have been few studies on the utilization of two kinds of coupling agents for the preparation of superhydrophobic coatings that may generate synergistic effects and result in the enhanced performance of the coating.

In this work, $SiO_2$ nanoparticles with hydrophobic properties were selected as the main source of the hydrophobic groups of the coating; epoxy resins with excellent mechanical properties were selected as the adhesives and 1,2-Bis (triethoxysilyl) ethane (D26) and (3-glycidyloxypropyl) trimethoxysilane (KH560) with different functional groups were selected as the coupling agents, which also provided low-surface-energy functional groups. The slurry was prepared by a simple blending method first, and then the superhydrophobic coating with high light transmittance and good durability was prepared by using the dip-coating method. Interestingly, the introduction of the two coupling agents not only enhances the long-term outdoor durability but also endows the coating with an excellent anti-icing and anti-frosting performance, much better than the coatings only using D26

or KH560 as coupling agent. The SiO$_2$@E51@KH560@D26 coating can also be realized on wood and aluminum surfaces with brush coating and spraying methods, suggesting that it is feasible for large-scale coating. Our work provides a fluorine-free and transparent SiO$_2$@E51@KH560@D26 superhydrophobic coating which shows both excellent anti-icing and anti-frosting performance and is feasible for large-scale application.

## 2. Materials and Methods

### 2.1. Materials

All reagents were used as received without further purification. 1,2-Bis(triethoxysilyl) ethane (D26), nano fumed silica (7-40 nm, 100 m$^2$/g), tetrahydrofuran (THF), and isopropyl alcohol (IPA) were obtained from Macklin (Shanghai, China). Bisphenol-A-based epoxy (E51, AR grade) was purchased from Aladdin Chemical Reagents Ltd. (Shanghai, China). The curing agent (T31) and (3-glycidyloxypropyl) trimethoxysilane (KH560) were obtained from Yousuo Chemical Technology Co., Ltd. (Shandong, China). Glass substrates were purchased from Shitai Experimental Equipment Co., Ltd. (Shitai, China) and were cleaned ultrasonically with ethanol for 15 min and dried prior to use.

### 2.2. Preparation of SiO$_2$@E51@KH560@D26 Superhydrophobic Coatings

The process for preparing SiO$_2$@E51@KH560@D26 superhydrophobic coatings is illustrated in Figure 1. Initially, a mixed solution consisting of deionized water (5 wt%) and isopropanol (95 wt%) was prepared and stirred magnetically. The pH value of the solution was adjusted to approximately 5 using acetic acid. Subsequently, D26 and KH560, with a mass fraction of 2%, were individually hydrolyzed in the mixed solution to obtain two different hydrolysis solutions. Next, 0.2 g of SiO$_2$ nanoparticles was dispersed in a hybrid solution comprising 7.5 mL of tetrahydrofuran (THF) and 7.5 mL of isopropyl alcohol (IPA). The dispersion was then subjected to treatment with a cell disintegrator for 10 min. Afterward, 0.25 g of E51, 0.5 g of D26, and 0.5 g of KH560 hydrolysis solutions were added to the SiO$_2$ dispersion. Additionally, 0.075 g of T31 was introduced, and the mixture was magnetically stirred for 2 h to form the coating solution. Finally, the SiO$_2$@E51@KH560@D26 coating was fabricated using a dip-coating method, followed by drying at room temperature for 12 h. For simplicity, the resulting coating was denoted as "E". To investigate the synergistic effects of the two coupling agents (D26 and KH560), SiO$_2$@E51@KH560 coating (referred to as "M") without D26 and SiO$_2$@E51@D26 coating (referred to as "N") were also prepared. The structural formulas of D26, KH560, and E51 are presented in Figure 2.

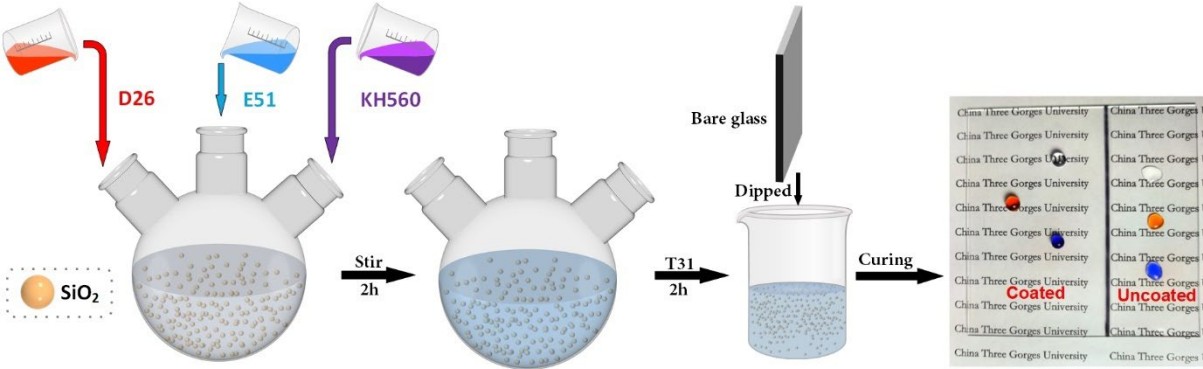

**Figure 1.** Illustration of the preparation procedure of SiO$_2$@E51@KH560@D26 superhydrophobic coatings.

**Figure 2.** Structural formulas of D26, KH560, and E51.

## 2.3. Materials Characterization

The microstructure of the prepared coatings was observed using scanning electron microscopy (SEM, JSM-7500F, Japan Electronics Co., Ltd., Tokyo, Japan). Fourier transform infrared spectroscopy (FTIR spectroscopy) was performed using a Fourier transform infrared spectrometer (NEXUS, Thermo Nicolet Corporation, Madison, WI, USA) in the wavenumber range of 800 to 3500 cm$^{-1}$ to identify the functional groups present in the coatings. The transmittance of the samples was measured using a UV-2550 spectrophotometer in the wavelength range of 300–900 nm. It should be noted that since the dip-coating process involved the deposition of coatings on both sides of the glass substrate, the transmittance results represent the combined effects of the coatings on both sides of the glass. The water contact angle (WCA) and water sliding angle (WSA) were measured using the Sci3000F Contact Angle/Interface system (Zhongchen Digital Technology Instrument Co., Ltd. Beijing, China). The volume of water droplets used for WCA and WSA measurements was 5 μL.

## 2.4. Mechanical Robustness, Chemical Stability, UV Robustness, and Self-Cleaning Performance Test

In the droplet impact test, the sample was positioned 20 cm beneath the faucet, and the droplet's descent velocity was calibrated to 1 drop per second. Subsequently, the WCA and WSA of the sample were gauged every two hours. For the sand and gravel impact experiment, 20 g of quartz sand, ranging from 0.18 to 0.4 mm in size, was released from a height of 40 cm to impact the surface of the sample. Following each impact, the WCA and WSA were reassessed. In the knife scratch test, scissors were employed to disrupt the coating, followed by the application of deionized water to verify its superhydrophobic characteristics. For the sandpaper wear test, a 100 g weight was placed on the sample, and it was then moved horizontally across 400-mesh sandpaper (Figure S2a). The WCA and WSA were measured after each wear cycle. In the tape-stripping test, a tape was pressed onto the coating under a load and then peeled off [27]. To assess chemical stability, the coating was exposed to acidic and basic environments with pH values of 1 and 11, respectively, for 10 days, with WCA and WSA measurements taken every 48 h. The UV durability of the coating was evaluated using a UV lamp operating at a power of 20 watts for 60 days. Throughout the test, the sample was kept 20 cm below the UV lamp, with WCA and WSA measurements conducted every five days. Prior to testing, all samples were washed with deionized water and dried in an oven at 40 °C for 30 min. Lastly, in the self-cleaning performance test, soil collected from a garden served as the contaminant to demonstrate the sample's self-cleaning capabilities. Initially, the soil was evenly distributed on the sample. Subsequently, continuous water droplets stained with methyl orange were poured onto the surface. Observations were made to assess the ability of the water droplets to effectively carry away the contaminant from the surface.

*2.5. Anti-Icing and Anti-Frosting Performance Test*

The anti-icing performances of the samples were rigorously evaluated by quantifying the icing delay times exhibited by water droplets and assessing the adhesive strengths between the formed ice beads and the sample surfaces. Furthermore, the anti-frosting properties of the samples were reflected through the meticulous observation of the frost formation processes, involving a comparative analysis of the condensate water's states and the quantities of frost accumulated.

### 2.5.1. Delay Icing Test

The icing process was conducted on a cooling table. Each sample was placed on the cooling table with 20 μL of deionized water droplets. Subsequently, the cooling table was activated until all the water droplets formed ice. The entire process was recorded using a single camera. Throughout the tests, the ambient temperature was maintained at $20 \pm 0.5$ °C, while the relative humidity level was maintained at $64 \pm 5\%$.

### 2.5.2. Adhesion Strength Measurement

The ice adhesion of the coatings was evaluated by a dynamometer which was purchased from HANDPI Instruments Co., Ltd. (Yueqing, China). The coatings were placed in a freezer at a temperature of approximately $-20$ °C, and 100 μL of deionized water was dispensed onto each sample. Once the water droplets froze completely, a push/pull meter with a digital display was employed to horizontally push the ice until it detached. The average value of the deionized water volume was calculated based on three experimental trials. The highest recorded force during the experiments was defined as the minimum deicing force/ice adhesion.

### 2.5.3. Ice and Thaw Cycle Test

To further analyze the anti-icing properties of the samples, icing and thawing cycle tests were conducted. In each cycle, 400 μL of deionized water droplets was applied to each sample and maintained at a temperature of $-20$ °C. After the deionized water had completely frozen, the samples were removed and placed at ambient temperature for a duration of 2 h to facilitate ice melting. For precise and convenient measurement, the back of the samples was marked. In each cycle, the water droplets were applied to the marked areas, and the WCA and WSA were measured on the marked regions.

### 2.5.4. Anti-Frost Test

The coatings were placed on the cooling table inclined at an angle of 4°. Subsequently, the cooling table was activated and the condensation and freezing of water vapor were observed. The entire process was recorded using a single camera. The ambient temperature was maintained at $23.5 \pm 0.5$ °C, while the relative humidity was $69 \pm 5\%$.

## 3. Results and Discussion

### 3.1. Morphological and Structural Characterizations

SEM images of E, M, and N coatings are presented in Figure 3a–i, revealing similar micro/nanostructures in all samples. These structures are the result of agglomeration of $SiO_2$ nanoparticles facilitated by D26 and KH560. The original $SiO_2$ nanoparticles underwent agglomeration, forming particles with an average size of approximately 100 nm, which further aggregated into larger particles measuring several micrometers in size. This consistent architecture ensures that the coatings possess superhydrophobic properties, as evidenced by the insets in Figure 3a–c and Videos S1–S3. These insets demonstrate that the WCA of the samples exceeded 156°, while the WSAs were below 1°. Figure 3g–i illustrate the thickness of coatings E, M, and N in this study. It should be noted that due to the uneven edge in the sample preparation process for these coatings, a rough estimation indicates that the thickness of the three coatings (composed of five layers) falls within the range of 0.35 to 0.5 μm. FTIR analysis was conducted to examine the chemical groups present

on the surface of the coatings. The corresponding results are shown in Figure 4a,b, with enlarged plots within the range of 1500~3900 cm$^{-1}$. Pure SiO$_2$ powder was also tested for comparison. In the SiO$_2$ FTIR spectrum, the absorption peak at 1068 cm$^{-1}$ can be attributed to the anti-symmetric stretching vibration of Si–O–Si [28]. The bands within the range of 3100–3700 cm$^{-1}$ represent the stretching vibration of the −OH group on the surface, while the peak at 1633 cm$^{-1}$ is attributed to the bending mode of water's –OH group [29]. In the case of the E, M, and N samples, the band originating from the –OH group and the peak at 1633 cm$^{-1}$ disappear due to the reaction between SiO$_2$ and coupling agents. Furthermore, two absorption peaks at approximately 2966 and 2913 cm$^{-1}$, corresponding to the symmetrical and anti-symmetrical stretching modes of C–H groups, were observed in the E, M, and N coatings. This confirms the presence of the –CH$_3$ group resulting from incomplete hydrolysis of KH560 and D26, enhancing the coatings' surface energy and superhydrophobic properties [25]. Additionally, the C=C group at 1510 cm$^{-1}$ and the absorption peak at 908 cm$^{-1}$ can be attributed to the aromatic and oxygen rings present primarily in the epoxy resin [30,31].

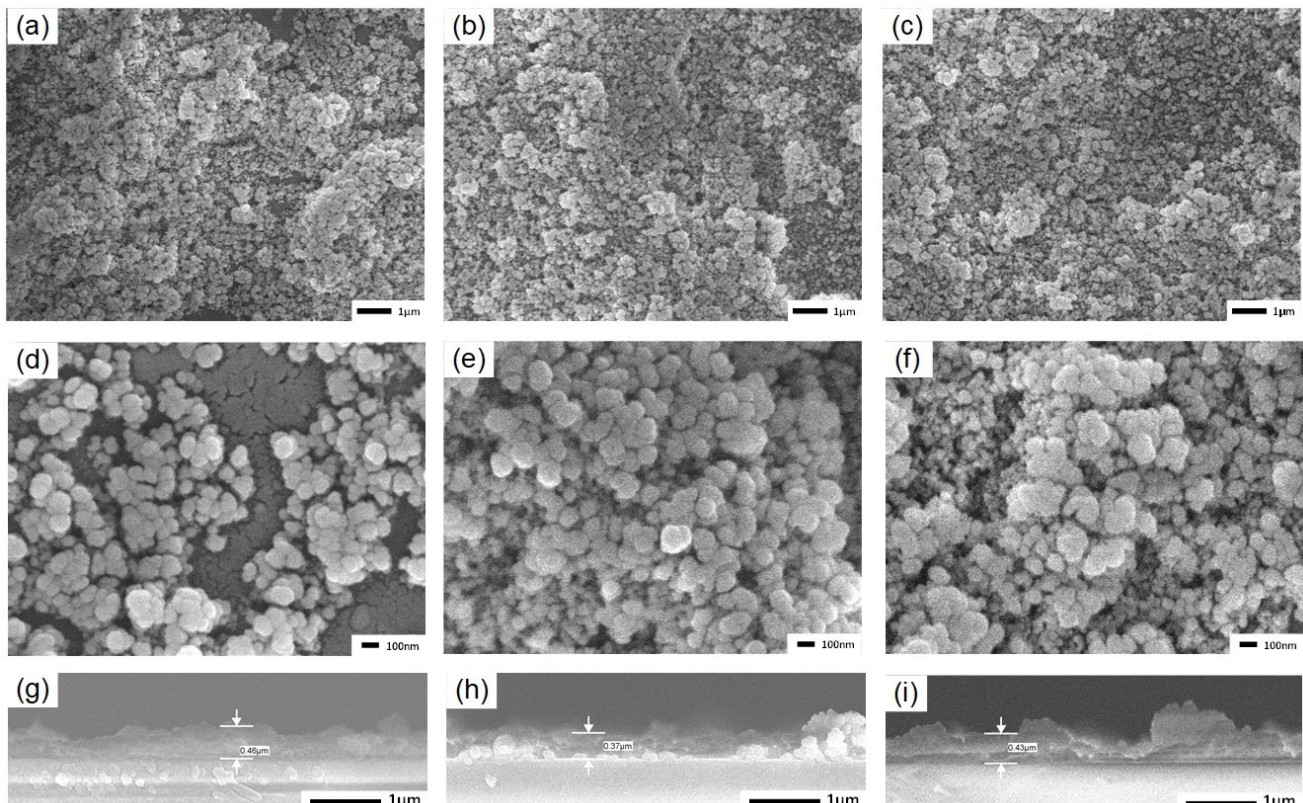

**Figure 3.** The microstructure in the coating SEM images of E (**a**,**d**), M (**b**,**e**), and N (**c**,**f**). The thickness in the coating SEM diagram of E (**g**), M (**h**), and N (**i**).

The transparency of the E sample was investigated using UV-Vis spectroscopy in this study. The transparency can be adjusted by varying the number of dip-coating layers. The scatter plot in Figure 4c presents the light transmittance across the entire wavelength range (300–900 nm) for films with different numbers of dip-coating layers. The average transmittance was calculated by averaging values within the recoverable band (400–750 nm). From the results, it can be observed that the bare glass exhibited an average transmittance of 91.7%. Surprisingly, a single dip-coating layer resulted in a significantly high transmittance of 86.3%. Although the average transmittance decreases as the dip-coating layer number increases, a transmittance of 80.2% can still be maintained when the dip-coating layer number is increased to five. However, when the dip-coating layer number is further

increased to 20, the average transmittance drops to 64.3%. It is important to note that the WCA did not change significantly with the increase in dip-coating layers.

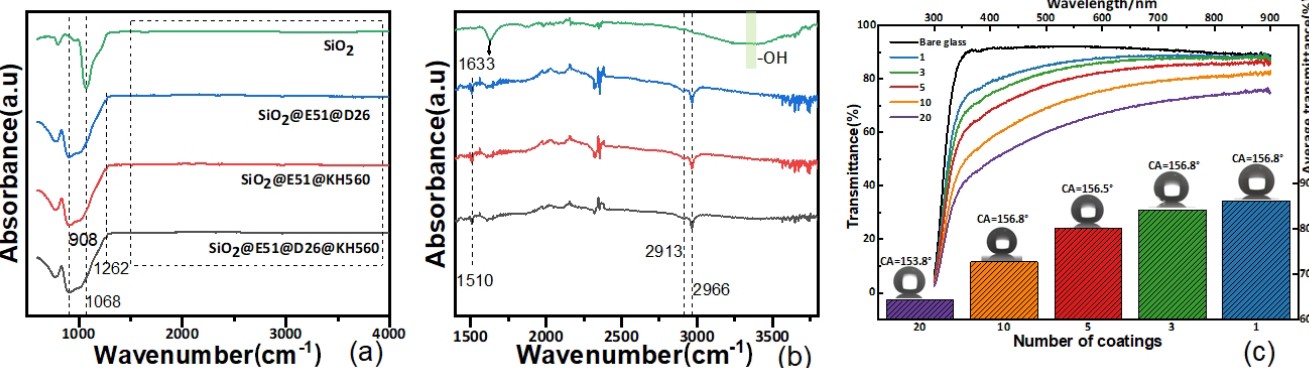

**Figure 4.** (**a**) FTIR spectra of the E, M, N, and pure SiO$_2$ and (**b**) the corresponding enlarged curves in the range of 1500~3900 cm$^{-1}$ for the E, M, N, and pure SiO$_2$. (**c**) Transmittance and WCA of the E coating with different numbers of dip-coating layers.

*3.2. Reaction Mechanism Analysis*

The possible reactions involved in the preparation of the E coating are illustrated in Figure 5. Figure 5a demonstrates the hydrolysis mechanism of a methoxy group in the D26 or KH560 molecules. It should be noted that both D26 and KH560 molecules contain multiple methoxy groups directly attached to carbon atoms, which readily undergo hydrolysis under acidic conditions, resulting in the formation of several hydroxyl groups [32]. Notably, each D26 molecule possesses six alkoxy groups, twice the number found in KH560. Consequently, the hydrolysis process of D26 may generate a greater number of alkoxy groups compared to KH560. Additionally, the silanol produced from the hydrolysis of D26 is more acidic compared to typical silane coupling agents. As a result, it can form stronger covalent bonds with surface hydroxyl groups on metals and inorganic materials, while exhibiting reduced susceptibility to further hydrolysis.

The epoxy resin curing agent, T31, employed in the experiment is a phenolic amine curing agent that has been modified through a Mannich reaction using phenols, aldehydes, and amines [33]. This particular curing agent comprises primary and secondary amine functional groups, as well as a phenolic hydroxyl group. These active hydrogen species can initiate ring-opening reactions with epoxy groups, leading to the generation of hydroxyl groups (Figure 5b–d). The majority of epoxy groups originate from the epoxy resin, while a smaller fraction is derived from KH560. These generated hydroxyl groups can dehydrate with the hydroxyl groups present on the surface of glass or SiO$_2$, thereby establishing stable bonds between the coating and the substrate (Figure 5e,f). Moreover, this curing agent can undergo condensation reactions with silanol or its alkoxy groups, facilitating bonding between the coating and the substrate. Additionally, self-condensation can occur among epoxy resin molecules (Figure 5g). The micro/nanostructure of the coating primarily arises from the polymerization between superhydrophobic silica particles and the coupling agent. A small amount of residual hydroxyl groups on the surface of these superhydrophobic silica particles, as well as a significant number of hydroxyl groups on the glass substrate surface, can interact with D26 and KH560, as depicted in Figure 5h,i [34]. Furthermore, it should be noted that the hydrolysis process of silane coupling agents is reversible [35] and, therefore, some methoxy groups may persist, contributing to the low surface energy and enhancing the hydrophobicity of the coating.

**Figure 5.** Possible reactions in the synthetic route. (**a**) Hydrolysis of coupling agents, (**b**–**d**) ring-opening reactions with epoxy groups, (**e**–**g**) condensation reactions, (**h**) reaction of hydroxyl groups on glass surface with coupling agents, (**i**) reaction of hydroxyl groups on $SiO_2$ with coupling agents.

### 3.3. Durability

Long-term outdoor durability, mechanical robustness, and chemical stability are crucial factors for the practical application of superhydrophobic coatings, as they are typically used in outdoor environments that are complex and harsh. In the long-term outdoor exposure tests, samples E, M, and N were directly exposed to natural outdoor conditions for 2 months (from 30 May to 31 July). The WCA and WSA of the samples were tested approximately every 5 days, and the results are presented in Figure 6a. The weather conditions during the test are described in Table S1. The error bar represents the standard deviation based on three measuring points. During the first 30 days, there was no significant change in WCA or WSA for all three samples. Interestingly, during the second 30 days, samples E and M still showed no obvious decrease in WCA (~155.0°), while the WCA of sample N decreased to about 151.0°. More importantly, the WSA of sample E remained around 2.0° throughout the entire test, while that of samples M and N increased to about 9.0° and 16.9°, respectively. These results indicate that sample E has the best weathering resistance among

the three samples. We analyze the potential reasons for this situation. Firstly, D26 can form stronger covalent bonds with surface hydroxyl groups on metals and inorganic materials, making it less prone to further hydrolysis. Secondly, KH560 provides a lower surface energy for the coating, thereby enhancing its hydrophobicity and reducing the erosion by water droplets during the test, resulting in higher durability. In summary, the combined effect of adding these two coupling agents contributes to sample E's superior weather resistance. Based on these findings, sample E is considered more suitable for long-term outdoor applications. The mechanical robustness and chemical stability of sample E were further investigated to demonstrate its potential application prospects.

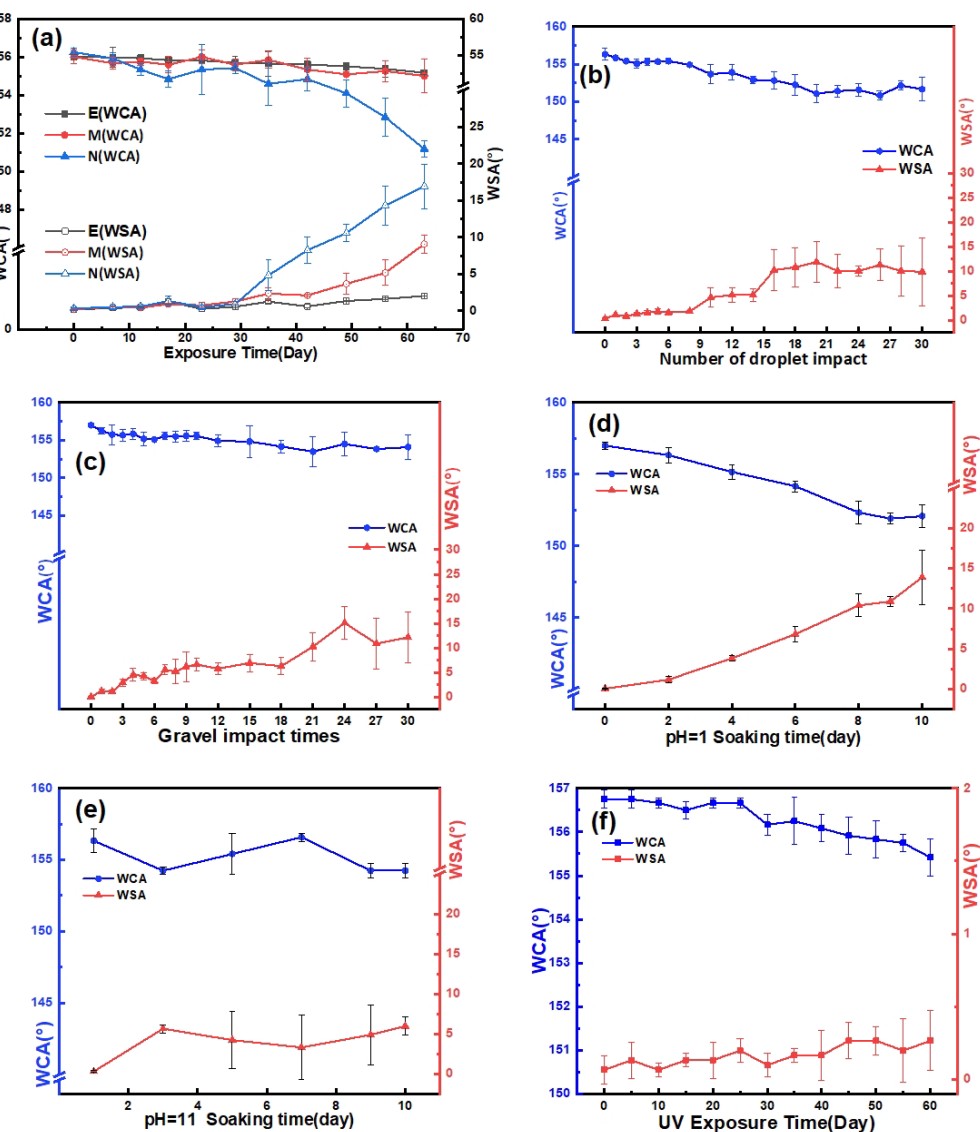

**Figure 6.** (**a**) The change in WCA and WSA of E, M, and N during weathering test. (**b–f**) The change in WCA and WSA of E coating after (**b**) sand impact test, (**c**) water droplet impact, (**d**,**e**) being immersed in corrosive liquids, and (**f**) UV radiation.

The mechanical robustness of sample E was evaluated through water droplet impact and sand and gravel impact experiments. Figure 6b,c demonstrate that the WCA of sample E remained above 150° even after 30 h of water droplet impact or 30 gravel impacts. In the knife scratch test (Video S6), although the coating surface exhibited scratches, it maintained stable superhydrophobic properties with a WCA of 156° and a WSA below 2°. Additionally, in the sandpaper wear experiment, the WCA remained above 150° even after undergoing

10 cycles of wear (Figure S2b). However, due to the thin nature of the coating, it may be partially stripped off the substrate and lose its hydrophobicity in the tape-stripping test. Regarding chemical stability, although there was a decrease in WCA and an increase in WSA over time, the coating retained its superhydrophobic characteristics with a WCA greater than 152° after being soaked in pH 1.0 and pH 11.0 solutions for 10 days (Figure 6d,e). The slight decline in performance could be attributed to the partial hydrolysis of Si–O–Si groups into Si–OH groups caused by the corrosive nature of the solutions during the long-term exposure [25,36].

The UV durability of the E coating was further assessed using a 20 W UV lamp for a duration of 60 days. The sample was positioned 20 cm below the UV lamp during the test. As shown in Figure 6f, after 60 days of UV irradiation, the total ultraviolet radiation dose accumulated to 23,328 J/cm$^2$. Throughout the entire test period, the WCA remained around 155.5°, and the WSA remained below 0.5°. These results indicate that there was no significant decrease in the hydrophobicity of the coating. The excellent UV resistance and aging resistance of the SiO$_2$@E51@KH560@D26 (E) coating can be attributed to its surface composition. The crosslinking bonds present in bisphenol A epoxy resin demonstrate strong stability and considerable UV resistance. Furthermore, SiO$_2$, with its wide band gap, is less susceptible to the effects of ultraviolet radiation [37]. These properties contribute to the E coating's remarkable long-term outdoor durability, mechanical robustness, and chemical stability, ensuring its stability as a superhydrophobic coating for applications in anti-icing and anti-frosting fields.

### 3.4. Self-Cleaning Capability

Self-cleaning refers to the automatic removal of surface contaminants without additional energy input. Effective self-cleaning is crucial for practical applications of superhydrophobic surfaces in the fields of photovoltaic panels, windows, buildings, and so on. As observed in Videos S4 and S5, water droplets effortlessly rolled off the E coating, effectively eliminating contaminants on the surface. In contrast, when water droplets came into contact with bare glass covered with dust, they penetrated the dust and wetted the glass surface. This can be attributed to the distinct adhesion properties of water between the E coating and bare glass surfaces. The abundance of hydrophilic hydroxyl groups on the bare glass surface makes it highly attractive to water. Conversely, due to its exceptional superhydrophobicity, the E coating exhibits significantly lower adhesion to water, facilitating effortless movement of water droplets along with carried-away pollutants. Such exceptional non-wetting characteristics are of great importance in preventing icing/frosting on solid surfaces.

### 3.5. Anti-Icing and Anti-Frosting Performance

Generally, superhydrophobic surfaces with micro/nanostructures are more effective in preventing ice and frost compared to hydrophilic and hydrophobic surfaces. The underlying mechanism can be summarized as follows: Firstly, the elevated free energy barrier on superhydrophobic surfaces impedes the formation of condensation water and slows down droplet growth rate. Secondly, due to their large contact angle and small water sliding angle, water drops exhibit weak adhesion when in contact with these surfaces, allowing them to roll, jump, or fall off once they reach a certain size (critical size) [38]. Lastly, the Cassie contact between condensate droplets and superhydrophobic surfaces significantly reduces their actual contact area, leading to diminished heat transfer efficiency. This delay in heat transfer delays water droplet freezing and frost layer growth [39]. The ability to delay icing is a key parameter for evaluating ice suppression. The icing process can be divided into three stages: Heat exchange stage, quasi-adiabatic glow stage, and ice growth stage. Since the second stage is rapid, the focus is mainly on the first and third stages when assessing icing delay performance. Figure 7 illustrates the icing process of water droplets (20 μL) on E, M, N, and O (bare glass) samples. On the bare glass (O), the droplets began freezing at 285 s and completely froze at 345 s. On the M- and N-coated

glass, the water droplets began freezing at 796 s and 903 s, respectively, and completely froze at approximately 932 s and 1087 s, indicating that both the M and N coatings can delay icing on glass surfaces. The icing delay time on the coated glass is about 2.7 and 3.5 times longer than that on the bare glass. Remarkably, on the E-coated glass, the water droplet started freezing at 1608 s and completely froze at 1974 s, resulting in an icing delay time approximately 5.7 times longer than that of the bare glass.

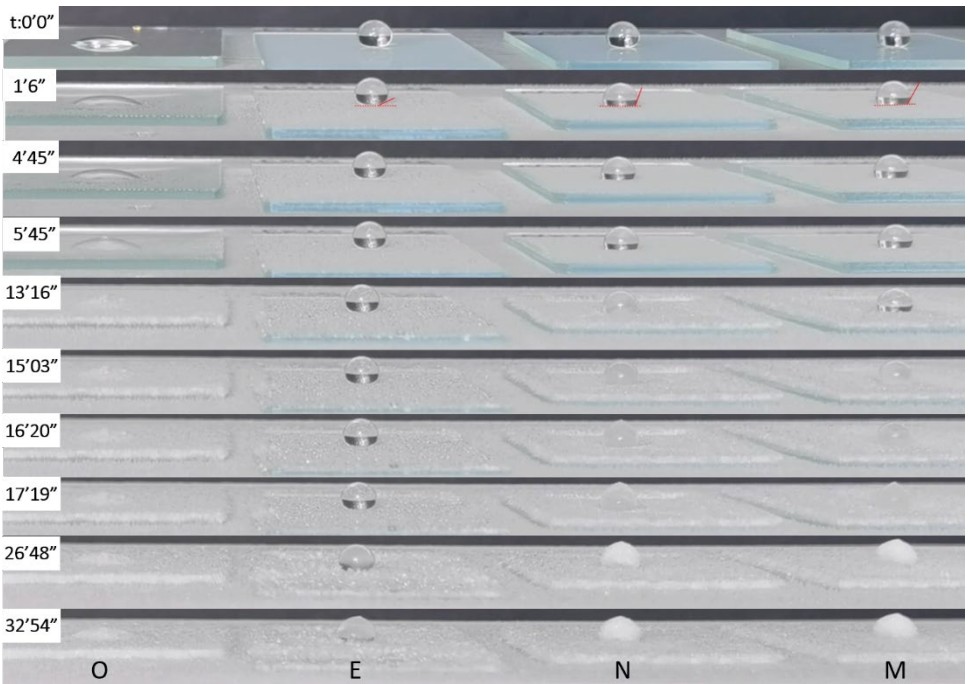

**Figure 7.** Freezing process of water droplets on O, E, M, and N samples.

Stefan et al. has reported that the both the roughness and wettability of the superhydrophobic surface are important roles on the icing delay time. When

$$x \leq \mathrm{O}(1), \tag{1}$$

$$f(R_a, \theta_{IW}) \approx f(R_a), \tag{2}$$

the freezing rate is mainly determined by the roughness of the substrate. When

$$x \geq \mathrm{O}(10), \tag{3}$$

$$f(R_a, \theta_{IW}) \approx f(\theta_{IW}), \tag{4}$$

wettability is the main factor in icing delay time, where x is defined as the ratio of the surface roughness parameter to the critical ice core radius (the minimum size required for the initial ice crystal to maintain a stable freezing process). Generally, the superhydrophobic surface satisfies Equation (3). According to previous studies [40], the wettability of a surface at freezing temperatures plays a crucial role in determining the rate at which freezing occurs. In this study, although the E, M, and N samples exhibited similar WCAs at room temperature, they showed significant differences in WCA during the icing process. As shown by the red line in Figure 7, even when the icing process began at 1608 s, the WCA of water droplets on the E sample remained above 155°. On the other hand, the WCA of water droplets on the M- and N-coated glass decreased rapidly during the icing process and reached approximately 114° and 120°, respectively, when the icing process began. This resulted in a shorter delay time for icing. This phenomenon can be attributed to the fact that as the WCA decreases, the contact area between water droplets and the coatings increases.

Similar observations have been reported in previous studies [41–43]. During the icing process, the temperature decreases, leading to an increased degree of supercooling, which refers to the temperature difference between the surface with steam and the cold surface. As a result, the bottom of the water droplet penetrates into the nanostructure, increasing the adhesion between the droplet and the matrix. This leads to a significant increase in the average and maximum diameters of the droplet. The water droplet gradually transitions into a piercing-Wenzel state, further increasing both the mean and maximum diameter. Eventually, the elliptical droplet completely penetrates into the nanostructure. According to the nucleation theory described in a previous report [44], a larger surface area undergoing supercooling reduces the energy barrier required for nucleation. Therefore, the larger WCA observed on the E coating during the icing process indicates that it maintains good wettability at low temperatures, likely due to its lower surface free energy.

The ice melting process was recorded and is displayed in Figure 8a. The cooling platform was adjusted to an angle of 4° from the horizontal surface. After the cooling platform was turned off, the ice on the four samples began to melt rapidly. On the bare glass, a large quantity of water remained on the surface after the ice melted. On the glass with E coating, the ice beads fell off quickly due to gravity before completely melting. At the same time, some small water droplets formed from the melting frost layer also rolled off immediately after merging. Interestingly, although the M and N coatings also exhibited ice delay capability, the molten water droplets struggled to fall off the glass, indicating that the hydrophobicity of the M and N coatings was significantly suppressed during the icing process and could not be recovered even when the temperature increased to room temperature. In practical applications, if the ice–water mixture cannot be separated quickly, it may freeze again and pose a risk to the anti-icing capability of the surfaces.

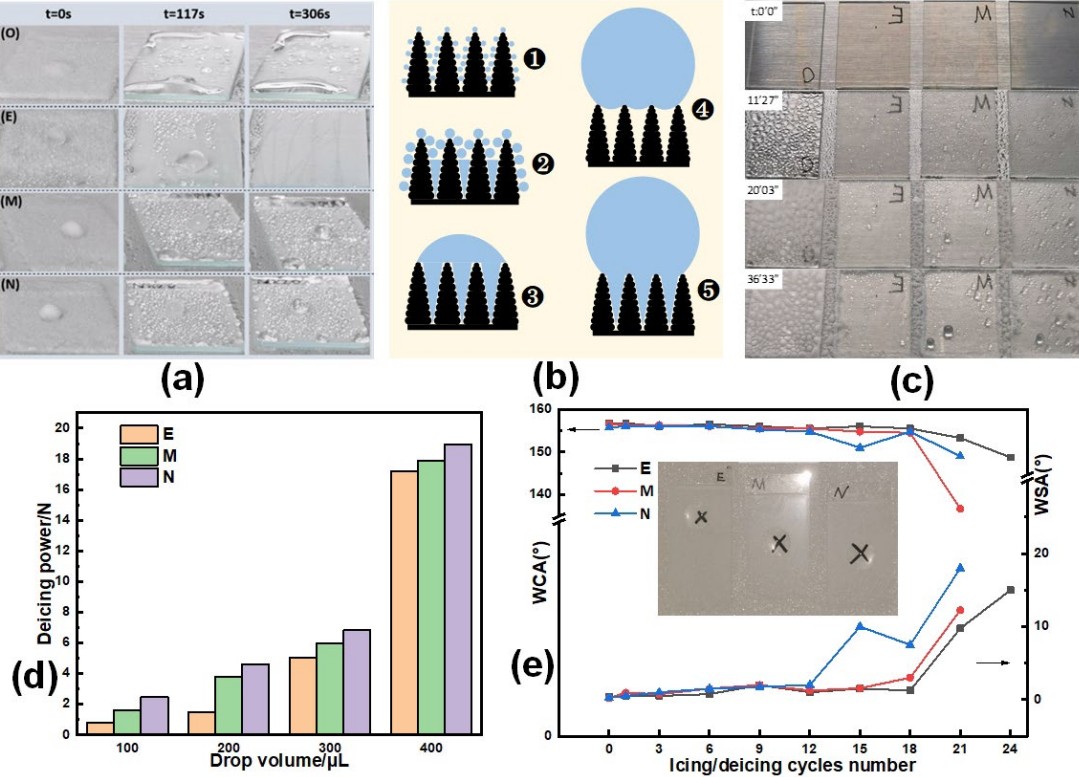

**Figure 8.** (**a**) Ice melting process on bare glass (O) and E-, M-, and N-coated glasses. (**b**) Diagram of condensing droplet condensation process. (**c**) Droplet condensation process on O, E, M. (**d**) Deicing force test. (**e**) The change in WCA and WSA of E, M, and N during icing/deicing test.

In the icing delay test, it was observed that the amount of frost accumulated on the surface of E was much less than the other three samples within the same time frame

(Figure 7), indicating better anti-frost performance. The defrosting process involves the condensation of water droplets on the surface during the initial stage of frost formation and the retention of molten water during the defrosting process. The formation process of condensate droplets is shown in Figure 8b (❶–❹ or ❶–❺), where ❹ indicates the Cassie state and ❺ indicates the Wenzel-puncture state. During the frost formation process, condensed water droplets freeze to form initial frost crystals. These frost crystals then develop into frost layers and continue to grow. Therefore, the condensation of water droplets serves as the foundation for frost growth [45].

In order to explore the frost suppression performance in more depth, anti-frosting experiments were conducted. On the bare glass surface, condensed water formed a continuous liquid film within approximately 12 min. Within 20 min, the glass became opaque, indicating that the liquid film had frozen. In contrast, on the E, M, and N coatings, water droplets were distributed evenly within about 12 min. After coalescence, these droplets spontaneously jumped and fell off the surface at a very high frequency (Figure 8c). The minimum radius of water droplets that can stabilize spontaneous detachment is referred to as the critical size. It can be observed that the diameter of the retained condensation water on the E coating was much smaller than that on the M and N coatings, indicating its superior anti-frosting performance. This is because the surface energy of the E coating, which determines the critical size for condensation droplets to roll, is much lower than that of the M and N coatings. These observations confirm the synergistic effect of D26 and KH560 in enhancing the anti-icing and anti-frosting performance.

Deicing force tests were also conducted in this study. Figure 8d shows the force required to detach deionized water, after freezing, from the E, M, and N coatings, respectively. According to the results, the ice detachment force on the same sample increased when the droplet volume was increased from 100 to 400 μL. However, for all samples, the force remained below 20 N when the droplet volume reached 400 μL. This suggests that even if ice has already formed on the surface, it can be easily removed by natural forces such as wind and gravity. It can also be observed that at the same droplet volume, the force required to detach ice beads from the coating follows the sequence E < M < N coating, indicating that ice is more easily removed from the E coating compared to the other two samples.

$SiO_2$-based superhydrophobic materials are widely used in anti-icing applications. In order to highlight the novelty of our work, we compared and summarized recent research on these materials (Table 1). It has been reported that certain materials can remove ice through photothermal [46–49] or electrothermal [50] processes. However, most of these materials are opaque. When compared to the transparent $SiO_2$@TCMS anti-icing material [51], the $SiO_2$@E51@KH560@D26 coating developed in this study exhibits higher light transmittance. Furthermore, it demonstrates excellent anti-frosting properties, which have not been extensively discussed in $SiO_2$-based anti-icing materials research. Additionally, fluorine-containing compounds are often added to improve the hydrophobic properties [47,49,52] of these materials, which can have a negative impact on the environment. In order to avoid this, we did not utilize any fluorine-containing compounds in the preparation of the $SiO_2$@E51@KH560@D26 coating.

To further investigate the durability of the superhydrophobic surface during the anti-icing process, we conducted ice-melting cycle tests in this study. In these experiments, droplets were subjected to icing and melting at the same predetermined position. As shown in Figure 8e, the WCA on sample E remained above 153°, and the WSA was below 10° after 21 icing and melting cycles. However, the WCAs on samples M and N decreased to 137° and 149°, respectively, while the WSAs increased to 12° and 18°. These results indicate that sample E exhibited higher durability in the anti-icing scenario compared to samples M and N.

**Table 1.** Comparison of recent anti-icing research.

| Superhydrophobic Surface | Light Transmittance | WCA | WSA | Anti-Icing Test | Anti-Icing Function |
|---|---|---|---|---|---|
| $SiO_2$/graphene composite [46] | Opaque | 153.0° | \ | Reduces ice-melting time by 387.5% | Photothermal deicing |
| Modified calcium sulfate whiskers@$SiO_2$-F/TPU [52] | \ | 159 ± 2° | 7 ± 1° | Prolongs the freezing time by 50 s | Delay freezing |
| EP@PDMS@graphene @$SiO_2$ [50] | \ | 164.1° to 157.0° | 2 to 4° | Electrothermal deicing/defrosting | Electrothermal |
| CNTs/PTFE-modified emulsified asphalt [47] | Opaque | 151.01° | \ | Freezing is inhibited under near-infrared light irradiation | Photothermal deicing |
| PDMS@$SiO_2$ and CNTs in silicon resin [48] | Opaque | 154.3° | \ | Delayed freezing time of 440 s at −20 °C | Photothermal deicing |
| $SiO_2$@TCMS [51] | 85% | 160.8° | 1° | Prolongs the freezing time by 40 min | Delay freezing |
| F-$SiO_2$/tea polyphenol/Fe NPs [49] | Opaque | 159° | <2.5° | The freezing time of surface droplets is extended by more than three times under 1 sun irradiation | Light absorption and photothermal conversion |
| $SiO_2$@E51@KH-560@D26 (this work) | 86.3% | 156° | 1° | The delayed freezing time is 5.7 times that of bare glass | Delays freezing, inhibits frosting, and accelerates deicing |

## 4. Conclusions

In this study, we successfully prepared a transparent $SiO_2$@E51@KH560@D26 superhydrophobic coating through a simple method. The WCA of the coating reached 156°, and the WSA was approximately 1°. Under the same experimental conditions, the freezing time of droplets on the coating surface was 6.1 times longer than that on bare glass. Additionally, it was 2.7 times longer than that of samples with KH560 alone, and 2 times longer than that of samples with D26 alone. These results indicate improved durability and low-temperature resistance of the coating. The improved performance can be attributed to the incorporation of D26 and KH560 into the coating. This not only enhances the durability of the coating but also ensures its superhydrophobicity even at low temperatures. A series of tests confirmed that the coating exhibits exceptional mechanical durability, acid/alkali resistance, and aging resistance. Furthermore, it outperforms coatings containing only D26 or KH560 in terms of delaying icing, reducing ice adhesion strength, and preventing frosting. This study highlights the significant potential application of the developed superhydrophobic coating in ice and frost suppression.

**Supplementary Materials:** The following supporting information can be downloaded at: https://www.mdpi.com/article/10.3390/pr12040654/s1, Figure S1: Adhesion strength measurement device schematic diagram; Figure S2: Sandpaper abrasion test; Table S1: Weather conditions during the weather resistance test; Video S1: E-WSA; Video S2: M-WSA; Video S3: N-WSA; Video S4: selfclean-E; Video S5: selfclean-bare glass; Video S6: knife scratch test; Video S7: selfclean-silicon dioxide; Video S8: selfclean-alumina; Video S9: selfclean-ferroferric oxide; Video S 10: wood; Video S11: Al.

**Author Contributions:** Conceptualization, T.X. and Y.W.; methodology, Y.X.; software, X.L. (Xinyi Li); validation, Y.W., X.L. (Xia Lang) and S.C.; investigation, Y.X.; resources, X.T.; data curation, F.T.; writing—original draft preparation, Y.W.; writing—review and editing, T.X.; visualization, Y.W.; supervision, T.X. and L.J.; project administration, X.T.; funding acquisition, T.X. and X.T. All authors have read and agreed to the published version of the manuscript.

**Funding:** This work is supported by the National Natural Science Foundation of China (No. 52007104), the Major Technological Innovation Project of Hubei Science and Technology Department (No. 2019AAA164), and the 111 Project (D20015) of China.

**Data Availability Statement:** Data are contained within the article and Supplementary Materials.

**Conflicts of Interest:** The authors declare no conflicts of interest.

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
