# Peer review of "Fluorine-Free and Transparent Superhydrophobic Coating with Enhanced Anti-Icing and Anti-Frosting Performance by Using D26 and KH560 as Coupling Agents"

_processes, doi:10.3390/pr12040654_

Round 1

Reviewer 1 Report

Comments and Suggestions for Authors

Having thoroughly read the article titled " Fluorine-free and transparent superhydrophobic coating with enhanced antiicing and antifrosting performance by using D26 and KH560 as coupling agents" I found it to be highly insightful and relevant. However, I recommend that the publication of this work be considered after the authors address the following significant details:

1.     Authors are encouraged to clarify the novelty of their work, considering the abundance of superhydrophobicity based anti-icing  studies published. Therefore, it is advised for authors to include a table summarizing recent typical research on the anti-icing performance of superhydrophobic coatings, illustrating the relevance of their work in this context.

2.     The introduction requires improvement. A recent important review articles are suggested to improve the introduction and add it to  the reference : Role of chemistry in bio-inspired liquid wettability. Chemical Society Reviews 51 (13), 5452-5497.

3.     Various crucial durability tests need to be conducted to ascertain the durability of the coating, including sandpaper abrasion tests, knife scratch tests and tape peeling tests. Authors are advised to refer to and cite the relevant article for insights on this matter: Porous and reactive polymeric interfaces: an emerging avenue for achieving durable and functional bio-inspired wettability. Journal of Materials Chemistry A 9 (2), 824-856.

4.     What is the thickness of these coatings?

5.     In addition to soil, is it possible for fine dust to self-clean on this superhydrophobic surface?

6.     Is this process surface-dependent? Is it possible to coat substrates other than glass with different characteristics?

7.     Dip coating is not feasible for large-scale coating. Is it possible to achieve spray coating without altering its superhydrophobic properties?

8.     Is this material able to repel liquids with surface tension lower than that of water?

Comments on the Quality of English Language

Minor editing of English language required

Reviewer 2 Report

Comments and Suggestions for Authors

This manuscript presents the effect of coupling agents on the durability, anti-icing and anti-frosting performance of the hydrophobic coating. Anti-icing and anti-frosting characteristics are indeed crucial for certain industrial sectors to maintain the performance of the equipment in low-temperature environments. The key findings of this work are interesting, and the overall idea of this work seems to be the original contribution to the field. This manuscript has the potential to be accepted for publication, but revision is required to improve the manuscript quality.

1.       There are several key findings obtained from the experimental investigations, as can be seen in the manuscript. Please concisely add more key findings in the abstract.

2.       On page 2; line 85, ‘enhanced the’ should be ‘the enhanced’.

3.       On page 3, the novelty/innovative aspects are not clearly stated in the introduction. Please emphasize the novelty/innovative aspects of this work in the last paragraph of the introduction.

4.       On page 3; lines 98 – 112, these paragraphs should not be included in the manuscript.

5.       On page 3, please change the title of section 2.1 from ‘chemicals’ to ‘materials’.

6.       On page 6, please separate the SEM images from the FTIR results.

7.       On page 8, please move Figure 5 to section 3.3.

8.       On page 12, Figure 7 (e) should be modified. It’s hard to identify which line represents the WSA and WCA.

9.       On page 13, please include some major quantitative findings in the conclusion.

Round 2

Reviewer 1 Report

Comments and Suggestions for Authors

Thank you for your response. After thorough revision, this article is now suitable for publication.

Reviewer 2 Report

Comments and Suggestions for Authors

The authors have satisfactorily improved the revised manuscript based on the given comments. Therefore, I recommend the acceptance of the manuscript for publication.